# Animal-Assisted Education: Exploratory Research on the Positive Impact of Dogs on Behavioral and Emotional Outcomes of Elementary School Students

**DOI:** 10.3390/children10081316

**Published:** 2023-07-30

**Authors:** Riki Verhoeven, René Butter, Rob Martens, Marie-José Enders-Slegers

**Affiliations:** 1Institute of Ecological Pedagogy, University of Applied Sciences Utrecht, 3584 CH Utrecht, The Netherlands; 2René Butter Psychological Advice, Van Brakelstraat 101, 3012 XW Rotterdam, The Netherlands; rene.butter@rbpa.nl; 3Department of Psychology and Education, Open University The Netherlands, Valkenburgerweg 177, 6419 AT Heerlen, The Netherlands; rob.martens@ou.nl (R.M.); marie-jose.enders@ou.nl (M.-J.E.-S.)

**Keywords:** animal-assisted education, social-emotional development, attunement, dogs

## Abstract

For some students, school success is not a simple matter. A growing, innovative approach that supports students’ functioning at school is programs in which animals are involved in education. The involvement of animals, especially dogs, in education is known as animal-assisted education (AAE). A literature review of AAE indicated a positive influence of AAE programs on the quality of learning and social emotional development in children. This study explored whether AAE positively impacts the social and emotional outcomes of elementary school students aged between 8 and 13 years through mixed methods. The methods used were a survey and an observational study. The survey section of the study showed that students participating in the program with the dogs rated themselves, after the intervention period, significantly higher in terms of self-confidence and had a more positive score for relationships with other students after the intervention. As rated by their teachers, after the intervention period, students scored significantly higher in relation to work attitude, pleasant behavior, emotional stability, and social behavior. In the observational study, we analyzed the video material of students who participated in an AAE program with dogs. We concluded that all verbal and non-verbal behaviors of the students increased, except eye contact. The current study indicates future directions for theoretical underpinnings, improved understanding, and the empirical measurement of the underlying variables and mechanisms.

## 1. Introduction and Hypotheses

School dropout is a major worldwide problem in education [1,2]. Unesco [3] reports that 258 million children do not attend school. ‘Inclusive education’ is seen as a way to address this problem. The essence of this policy is to organize care for students with special needs as much as possible at home and within a regular school setting [4]. However, the successful implementation of inclusive education, to reduce drop out, is difficult. One reason is that many teachers suffer a lack of competence when dealing with the often complex and increasing problems of the students’ behavioral, cognitive, and emotional abilities [5].

Dealing with such problems may for instance require physical contact. Since touching students is becoming more and more restricted and has been protocolled (even more strictly since the COVID-19 pandemic), the educational task for teachers has become even more difficult. Bowlby [6], Moberg [7], and Manen [8,9] emphasized the importance of physical contact for children’s healthy development, including in educational environments. Unfortunately, until now, inclusive education did not meet all expectations when it came to solving school dropout problems. The challenge is how to motivate children to attend school and help children acquire better academic and life skills. 

Over the last two decades, many studies have investigated the effects of human–animal interactions, such as animal-assisted interventions. This is a relatively new and emerging field. The International Association of Human–Animal Interaction Organisations (IAHAIO) introduced in its white paper the term animal-assisted interventions (AAIs), defined as “goal-oriented and structured interventions that intentionally incorporate animals in health, education, and human service for the purpose of therapeutic gains and improved health and wellness” [10]. 

Concepts and frameworks to explain the effects and mechanisms of human–animal interactions are still being developed. One such framework is that of enactive anthrozoology [11], which is an attempt to develop a common ground for the underpinning of human–animal interactions (HAIs). Verheggen et al. [11] described a model in which frequently used theories are integrated to understand the relationship between humans and animals (see Figure 1). The basic idea of the model is to understand which theoretical principles influence the mutual embodied attunement between living systems. 

AAI as defined by IAHAIO comprises different forms. First, there is animal-assisted therapy (AAT), which consists of structured therapeutic interventions applied across a variety of disciplines and includes occupational psychologists, social workers, and educators. Another specific AAI is animal-assisted activity (AAA), which aims to encourage participants and bring recreational benefits, usually through informal arrangements, and is conducted by specially trained volunteers, professionals, or paraprofessionals. An innovative approach that involves the participation of animals in educational settings to promote learning and development among students is animal-assisted education (AAE).

In this special form of animal-assisted intervention (AAI) for children and young adults, a professional with a trained dog aims to develop academic, social-emotional, and cognitive functions in children’s education [12]. In addition to the play and learning elements, it contains reciprocal physical contact between the child and animal in a natural manner.

A literature review of AAE was conducted in [13] based on international studies in schools, including different populations of children and different research designs. Fifteen of the eighteen studies included in this review were conducted with dogs. The unique bond between humans and dogs can be attributed to their co-evolution over the centuries, allowing them to work together seamlessly, especially in play [14,15]. Emotional recognition is also crucial in human–dog interactions, as indicated by researchers [16,17]. Various features have emerged from research on human–animal interactions. Rooney and Bradshaw [18] discovered that a shared attention focus and observational learning in play are essential components of the human–dog relationship. Jalongo et al. [19] pointed out that dogs provide a secure basis and foster a reciprocal attachment bond, like human–human attachments. Communication is another important factor in human–dog interactions, relying on visual and social cues [20,21]. Physical and affective contact also plays a role in human–dog interactions [22,23].

Studies have shown increased motivation and reading achievement due to the presence of a dog [24,25,26]. Barber and Proops [27] concluded that motivation to read was significantly higher in the presence of a dog. Meanwhile, Trammell [28] concluded that interaction with dogs reduced stress in an exam situation. Elsewhere, Schretzmayer et al. [25] concluded in a study within elementary education that dogs have a physical calming effect. Social interaction between a student and teacher increases when AAE is applied [29]. Elementary school teachers perceived that children were more likely to share their emotions, which resulted in increased empathy and improved social-emotional functioning [30]. Dicé et al. [31] also concluded that relational skills increased. The concept of play appears to have an important role in education. In Connell et al.’s [32] study, ‘play’ was specifically mentioned. In addition, several studies from the literature review mentioned the casual, playful, and relaxed nature of the intervention [24,27,29,31]. According to Beetz et al. [33], dogs can operate as mediators between a child and teacher. There are indications that children who have difficulties in attuning to adults are, on the other hand, able to attune to animals [33].

An important challenge regarding the growing popularity of the participation of animals in the classroom/school is that the effects have not yet been sufficiently theoretically and/or empirically substantiated. Quite often it is mostly based on teachers’ intuitions, which are difficult to research [34]. This makes gaining insight into any effectiveness problematic. Despite this, there are indications that children benefit from programs involving animals [11,12,13,33,35]: for example, AAIs for dyslexia could increase self-esteem and improve cognitive functions and language skills recovery [36].

Despite this growing body of literature, much remains unclear. Verhoeven et al. [13] concluded that the relevant studies were not always comparable, used different designs and methods, and lacked a common theoretical framework. We need to identify the effects of AAE and generate ideas and hypotheses to enhance further research. To this end, we conducted this explorative study in the Netherlands in the period 2017–2021 within a group of collaborating elementary schools offering AAE as extra support to students aged between 8 and 13 years. More information about this extra support is described in Section 2.2. In the Netherlands, elementary education is compulsory for pupils aged 5 to 12 years. Due to certain circumstances, students aged 13 years or older sometimes continue elementary education. 

Based on the existing literature, our main question and hypotheses are presented below.

The main question can be specified as follows: does AAE positively impact the social- and emotional outcomes of elementary school students aged between 8 and 13 years?

**H1:** 
*AAE positively impacts the social and emotional outcomes of elementary school students aged between 8 and 13 years, as reported by the students themselves after the intervention (self-confidence, work attitude, relationship with the teacher, and relationships with other students).*


**H2:** 
*AAE positively impacts the social and emotional outcomes of elementary school students aged between 8 and 13 years, as reported by their teachers after the intervention (working attitude, pleasant behavior, emotional stability, and social behavior).*


**H3:** 
*Underpinning H1 and H2, AAE positively impacts the level of intensity and energy reflected in the verbal and non-verbal communication of students during the intervention.*


## 2. Survey Study

### 2.1. Study Design and Population

The quantitative part of the study had a quasi-experimental pre-test/post-test design without randomization. In research methodology, a quasi-experimental design refers to a study that lacks the random assignment of participants to different groups. Instead, pre-existing groups or naturally occurring conditions are used to study the effects of an intervention or treatment [37]. To enhance the validity of the quasi-experimental research, a control group was utilized to establish a reference point for comparison. The control group did not receive the intervention but underwent the same pre-test and post-test measurements.

In this study, which involved the embedding of the DOG Project as part of ‘De Driegang’, randomly assigning students was impractical. Thus, this quasi-experimental design provided an alternative approach to evaluate the intervention’s effectiveness. Despite the limitations inherent to quasi-experimental research designs, they play an important role in advancing the comprehension of cause-and-effect relationships within real-world contexts.

Questionnaires were employed to collect the data. The research design increased the plausibility that any results found could be attributed to the intervention [37].

To participate in the experimental group, students had to meet the following inclusion criteria: be aged between 8 and 13 years old, obtain a recommendation from the school to participate in the project, have a clear request for help made on their behalf from the school and parents/caregivers that goes beyond the basic care provided by the school, and have permission from their parents/caregivers. The request for help, depending on the student’s school concerns, had to pertain to providing support for students who require extra attention in the domains of social or emotional self-efficacy, the development of self-confidence, or the stimulation of communicative self-efficacy. The registration for the DOG Project followed the criteria formulated by De Driegang [38]. The control group consisted of a student matched by their classroom teacher with the student participating in the experimental group. The criteria for the control group were as follows: the student had to be a classmate of the student participating in the experimental group; there had to be a request for help; and, during the intervention period of the DOG Project, the student could not be enrolled in any other special program.

After obtaining permission from 37 parents/caregivers for their child’s participation in the experimental group of the study, 37 classroom teachers were approached to participate and complete the questionnaires. The teachers were also asked to facilitate the completion of questionnaires by the students in both the experimental and control groups. The students of the control group were identified and chosen by the classroom teacher. The students who were part of the control group remained anonymous to the researcher. However, 9 classroom teachers did not fill in the questionnaires after the permission of the 37 parents/caregivers was obtained, reducing the number of participating teachers to 28. As a result, there were 28 students in both the experimental and control groups. However, six students in the experimental group and five students in the control group provided incomplete or inoperable answers, leaving a total of twenty-two students in the experimental group and twenty-three students in the control group with completed and usable questionnaires after the intervention. The study was approved by the Research Ethics Committee of the Open University, Heerlen (approval date 16 December 2015, U2015/08468/HVM). 

Teachers and students completed Dutch Instrument for Social-Emotional Development questionnaires (known in the Netherlands as VISEON) [39]. The questionnaires were provided in paper format. The measurement times were before and after the completion of the DOG Project. It took approximately 30 min to complete each questionnaire.

VISEON monitors the social-emotional development of pupils in elementary education, firstly because the core objectives for elementary education make it clear that education must also focus on content that transcends the curriculum (decree on core objectives for elementary education, 1998), and the social emotional development of pupils is part of this, and secondly because there is a clear interaction between a child’s social-emotional development and his or her development in other areas. Problems in the social-emotional area can negatively affect the educational learning process. Equally, it is possible for learning problems to interfere with a child’s development in the social and affective areas. 

The independent variable in this study was the intervention of the DOG Project, while the dependent variables were the questionnaire-based measurements of the social emotional development of the participating respondents. 

### 2.2. Procedure

In the ‘De Driegang’ (Appendix B) partnership of schools, 45 schools work together, with their expertise center ‘De Rotonde’ offering extra support to students aged between 8 and 13 years old. Examples of this support are speech therapy, play therapy, working with a horse, and working with dogs [38]. The DOG Project was conducted by a specialized teacher/handler and trained dogs working together to provide intervention to students in a room near their classroom within the school [39]. Three trained male dogs participated during the period 2017–2021: a flat-coated retriever–golden retriever mix, a flat-coated retriever, and a flat-coated retriever–German shepherd mix. The dogs participating in the program were owned by the teacher/handler who carried out the program and was specialized and trained in working with students and with dogs. 

Prior to the start of the program, the involved classroom teacher and parents/caregivers of the student were approached for participation in the study and provided with an information letter detailing the study’s objectives. The letter addressed the data collection process and emphasized that one student from the experimental group and one student from the control group would be selected by the same classroom teacher. Parents/caregivers were required to sign a consent form for their child’s participation, with the option to withdraw at any point during the study. There were no dropouts during the study period. The Partnership took care of all procedural matters such as insurance. The program with dogs is an intervention aimed at improving the functioning of students during their education. The DOG Project consisted of ten weekly sessions for individual students (Appendix B). It started with an introductory meeting, followed by eight training sessions and a closing presentation. The first session served to introduce the program and the dogs to all classmates and the teacher. In the tenth session, the student presented in the classroom what they had achieved during the program. Each session consisted of parts with and without a dog. A teacher/handler and three dogs participated in the program. During sessions 2–9, the student invited a classmate to participate. Each session ran for an estimated 60 min, in which the dog participated for 15 to 20 min. During the week, multiple students participated in the program, which was conducted over four separate days. The program and the dogs were free during vacations (12 weeks a year). Each day, a maximum of two students could participate, and typically a dog took part in one session per day. 

The handler/teacher had a car specially equipped for the dogs’ transportation to the schools. Each dog had its own resting place inside where they stayed before and after the session. Each dog wore a vest indicating that they should not be petted without permission. During the part of the session involving the dog, the handler/teacher brought along a mat for the dog to lie on. Prior to each session, agreements were made and reiterated with the student regarding the interaction with the dog. The guidelines of IAHAIO for working with animals in AAI were followed: the wellbeing of the dog was closely observed and, under signs of stress, the session would end immediately.

### 2.3. Outcome Measures

To monitor and improve the cognitive and social emotional development of students, schools use a system that is evaluated by the Netherlands Test Affairs Committee [40]. The Instrument for Social-Emotional Development (VISEON) is part of this system and includes both a teacher observation list and a self-assessment list for students. Both lists measure scores from a Likert-type scale. The teacher list contains 44 questions formulated as pairs of opposing statements that relate to concrete observable behavior, focusing on four dimensions of social-emotional functioning: a careful working attitude versus a careless working attitude, pleasant behavior versus disruptive behavior, emotional stability versus emotional instability, and social behavior versus withdrawn behavior. The teacher list focuses on the direct and concrete observable behavior of students at school, analyzing the behavior expressed while dealing with school tasks, fellow students, and the teacher. The dimension of a careful working attitude versus a careless work attitude refers to the teacher’s perception of the student’s work behavior—his or her attitude toward learning at school. The dimension of pleasant behavior versus disruptive behavior relates to the teacher’s perception of the student’s behavior toward his or her classmates and toward the teacher himself or herself, as well as the extent to which the teacher believes the student can be considerate. The dimension of emotional stability versus emotional instability centers around the sense of security and steadiness in a student’s behavior, as perceived by the teacher. The statements contrast the confident, emotionally stable student with the insecure student who is easily upset. The social behavior versus withdrawn behavior dimension relates to the extent to which the student, according to the teacher, is focused on participating in group activities and interacting with the teacher and fellow students. On the one hand, it refers to the present, social learner who is open and self-confident and takes initiative in establishing contacts. On the other hand, there is the closed and withdrawn pupil who behaves timidly, prefers to remain in the background, and infrequently takes initiative. 

The Big Five personality structure is used in VISEON as the theoretical basis for the development of the teacher list [41]. The scales were developed using the one-parameter logistic model (OPLM) [42].

The student questionnaire focuses on the student characteristics underlying the behaviors that the teacher questionnaire also addresses: the behaviors expressed while dealing with peers, school tasks, and the teacher. The student questionnaire allows students to provide concrete information about how they perceive their own behavior in the social and affective spheres. The student questionnaire contains forty-two statements with four response options, assessing self-confidence, work attitude, relationship with the teacher, and relationships with other students. 

The self-confidence dimension indicates the extent to which the child is confident in his or her own abilities and is up to the task. A student’s self-confidence is determined by his or her self-image [40]. This dimension reveals whether a child is sensitive to criticism and how criticism affects his or her self-confidence. The relationship with the teacher dimension refers to the student’s attitude toward his or her teacher. This dimension reveals how the student experiences the teacher—whether the student feels that the teacher exerts a positive or negative influence on him or her. The relationships with other students dimension refers to the student’s attitude toward his or her classmates. Aspects of this dimension include whether the student likes or dislikes others, whether he or she dares to make contact with the other students, and whether or not the student recognized undesirable behavior from other students.

The experimental group in this study completed both student and teacher questionnaires on social-emotional development at the start and end of the intervention, while the control group completed the same questionnaires at the same time points, but without participating in the intervention.

### 2.4. Analysis

First, the results from the VISEON questionnaires of the students and teachers were coded and recoded where necessary and imported into an SPSS database file [43]. The one-tailed Wilcoxon signed rank test was used to measure differences in the values of the paired dependent variable. A non-parametric test was chosen because of the small sample size and lack of a normal distribution. In both the experimental and control group, results were examined pre- and post-test for the three separate dimensions in the student’s self-assessment (self-confidence, student relationships, and teacher relationship) and the four separate dimensions in the teacher’s assessment (attitude to work, pleasant behavior, emotional stability, and social behavior).

### 2.5. Questionnaire Results

The findings of the study were derived from the complete responses provided by 23 students in the control group and 22 students in the experimental group. Following the baseline measurement, the students in the experimental group individually underwent the DOG Project intervention, while those in the control group did not receive any intervention. To enhance the validity of the quasi-experimental research, a control group was utilized to establish a reference point for comparison. The control group did not receive the intervention but underwent the same pre-test and post-test measurements. By contrasting the changes observed in the treatment group with those of the control group, we aimed to isolate the effects of the intervention.

Because of our sample size and the non-normality of the data, we followed a non-parametric approach. The control group was used to show in a qualitative way that the passing of time or maturation probably did not explain the results of the experimental group. The data had to be analyzed separately for the experimental group (Table 1) and the control group (Table 2). We also report the results of the control group, because if a difference was also found between the pre-test and post-test results in the control group, the results in the experimental group were less trustworthy.

The VISEON results for the students’ perceptions in the experimental and control group were different.

When comparing the pre-test and post-test results of the experimental and control groups, it was found that students in the experimental group scored significantly higher on self-confidence in the post-test and more positively in the measurement of their relationships with other students. The experimental group also reported a higher score in the post-test regarding the perceived positive influence from the teacher as compared to the pre-test. The control group’s scores showed no significant changes between the pre- and post-test.

In the post-test, teachers scored the students in the experimental group significantly higher in all four dimensions (attitude to work, pleasant behavior, emotional stability, social behavior).

## 3. Observational Study

### 3.1. Study Design and Population

The observation study aimed to explore the impact of animal-assisted education (AAE) on verbal and non-verbal communication during a session. To accomplish this, 37 students were recorded by video during the fifth session of the protocolled program (see Appendix A). Each session lasted 60 min, and during each session approximately 25 min was recorded. The session’s topic was “play”, and the recordings were taken across various schools and locations. However, only recordings of 26 students in different schools were suitable for analysis due to differences in quality and structure. Two students were excluded because they had an incomplete session. 

The qualitative observational study was conducted by analyzing video material of the sessions. Observations on the behavior of the students were conducted in diverse (class)rooms in schools. Data were collected using non-random purposive sampling because the research question depended on a highly targeted, specifically defined population. In this case, the sample was limited to children in schools in the Netherlands with social-emotional learning problems. From these children, selections were made of students who were part of the DOG Project and whose video footage met the defined criteria as listed in Table A1 (in Appendix A). 

### 3.2. Procedure

The observational study was, as with the survey study, conducted in relation to the DOG Project, part of ‘De Driegang’ (Appendix B), involving students aged between 8 and 13 years old. The parents/caregivers of the student participating in the DOG Project were approached for participation in the study and provided with an information letter detailing the study’s objectives. Parents/caregivers signed a consent form for their child’s participation, with the option to withdraw at any point during the study. The Partnership took care of all procedural matters such as insurance. Furthermore, the procedure and content of the DOG Project outlined above are also applicable here.

### 3.3. Video Analysis

An observation list (Table A1, Appendix A) was used to score the videos on different aspects of verbal and non-verbal communication. The list covered elements such as facial expressions, eye contact, posture, voice volume, and physical contact with the dog. Three observers rated the videos to establish intercoder reliability [37] using the numbers 1 to 5 to indicate the degree to which a student demonstrated a particular aspect. The distinct aspects of the communication types were formulated through opposing indicators. For example, the aspect of voice volume was indicated by a score of 1 as soft and 5 as loud. A score of 2 was more soft than loud, a score of 4 was more loud than soft, and a score of 3 was neither soft nor loud (Appendix A). Time sampling was used, with recordings paused every ten seconds for observation. The study sought to ensure consistency by filming moments during game tasks, with the presence of the dog, the teacher/handler, and the filming researcher. Videos were chosen as the medium for recording as they allowed for repeated viewing, the detailed analysis of short frames, and verification by multiple researchers. The clips were chosen based on the abundance of usable footage, the quality of the footage, and whether they met the criteria. The criteria for the clips were chosen to eliminate as many potentially confounding variables as possible and to increase the homogeneity of the clips, trying to ensure that the only changing variables were the independent variables: (1) the student, (2) the presence of the dog, and (3) the presence of the teacher/handler. The following criteria were used: (a)The clips had to cover the same task (game).(b)The student, teacher/handler, researcher, and dog had to be present in the room.(c)The clips had to be taken from three parts of the intervention session (free situation with the dog, first task with the dog, and second task with the dog).(d)The clips had to show clear interactions with the student.(e)The clips had to be taken from the same day.

The video material was evaluated in terms of various aspects of communication based on the observation scheme of Foster [44] by three independent raters. A total of 229 video fragments were collected to evaluate the three conditions. Of these, 66 fragments were selected for the free situation, 81 for task 1, and 83 for task 2. The material for each student that met the criteria totaled 4 min for the three conditions. Each of these fragments was reviewed and scored by three researchers.

### 3.4. Video Statistical Analysis Results

#### 3.4.1. Statistical Analysis

The Fleiss Kappa coefficient was used to measure inter-rater reliability, and it was found to be 0.80, indicating satisfactory reliability. To obtain an aggregated score, the results of the three assessors were averaged, and their mean differences were examined using the one-sided Wilcoxon signed-rank test. The significance of the results was determined by calculating a *p*-value, which indicated the likelihood that the findings were due to chance. In this study, the conventional threshold of *p* < 0.05 was employed, which meant that there was a 5% chance that the results were obtained by chance when the null hypothesis was true.

The objectives of the DOG Project’s interventions were to elicit target behaviors and to promote adequate behavior. The level of task difficulty gradually increased from task 1 to task 2. The means were compared using the one-sided Wilcoxon signed-rank test (see Table 3). 

#### 3.4.2. Results

The impact of animal-assisted education (AAE) on verbal and non-verbal communication was noticeable for most behavioral aspects, with the average score increasing in various patterns between the free situation and task 1 and task 2. However, there was no apparent change in “eye contact” and “eye contact with the dog”, which could have been due to these aspects being challenging to assess or not occurring at all. Additionally, in the intervention with the animal and toy, we noticed higher scores. The higher score in “contact with dog and toy” was noteworthy, as the dog returned the toy to the child during the exercise. The higher score suggested that the child’s confidence and posture were growing during this activity. 

## 4. Discussion

AAE is a growing field of interventions in education, but it still heavily relies on educators’ intuitions. The present study provides insights for both the theoretical underpinning and an improved understanding of the underlying variables and mechanisms through empirical measurement. Future directions should focus on further developing the theoretical framework and conducting research to enhance the understanding and empirical evidence in this field. Regarding motivation, we could not derive a causal explanation from this exploratory study. The assumption was that, consistent with other studies [24,25,26], working with dogs increased motivation during activities. The increased intensity levels of the students’ behaviors that were observed during the intervention may indeed have been related to motivational aspects. Verhoeven, Enders, and Martens [13] described, based on self-determination theory [45], that goal-directed behavior, psychological development, and well-being can be achieved above all through intrinsic motivation. Studies have shown that children with social-emotional problems are quite often difficult to motivate [46]. Interestingly, the concept of intrinsic motivation comes quite close to the concept of play. Many researchers have stated that play is hard to define: “Play is probably one of the most misunderstood areas in relation to children’s education and development” [47]. Nevertheless, there is consensus that play is common in all (young) mammals as well as in humans, and that its primary function relates to learning and social development. It is beyond the scope of this text to define play in detail, but the definition of Gray [48] clearly demonstrates a strong overlap with intrinsic motivation, as he observed that “an activity can be defined as play, or as playful, to the degree that it is (1) self-chosen and self-directed; (2) intrinsically motivated; (3) guided by rules; (4) imaginative; and (5) conducted in an active, alert but relatively nonstressed frame of mind” [48]. Many researchers have also added to this that play serves no clear external purpose. The goal seems to be decided in the game itself, that is, in the process. The voluntary nature of play, the pleasure involved, the curiosity that often accompanies play, and the sense of self-direction show a strong overlap with the concept of intrinsic motivation from self-determination theory [45,49]. ‘‘Perhaps no single phenomenon reflects the positive potential of human nature as much as intrinsic motivation, the inherent tendency to seek out novelty and challenges, to extend and exercise one’s capacities, to explore, and to learn” [45]. The strong overlap between play and intrinsic motivation is confirmed by neurobiological research, since “affective neuroscience suggests that human intrinsic motivation is based in ancient mammalian systems that govern exploration and play” [50]. Playful interaction with dogs might increase intrinsic motivation, which, as shown in many (meta)studies, produces many positive effects, not only on self-regulation, emotional and social development, and deep-level learning, but also on a sense of well-being [45]. To conclude, the relationship between play and intrinsic motivation could provide a good opportunity to theoretically substantiate the complex effects of AAE.

According to the results of the survey and observational study, we endorse the impact of AAE with dogs, such as in the DOG Project, on engaged relationships and higher levels of autonomy. This is especially useful for students with social emotional learning difficulties who need “extra attention in the area of social-emotional independence”, “self-confidence development”, and the fostering of “communication and other life skills”, in which play-based learning is central [51]. It also seems to improve feelings of competence. Upon comparing the pre-test and post-test results, it appeared that the DOG Project had a beneficial influence on the social-emotional development and social behavior of students. The results from the observational study suggested that the program, in the context of mutual embodied attunement (i.e., the dynamic interaction between the student, dog, and professional teacher/handler), may be a helpful mechanism that also positively influences the development of communication. This finding highlights the potential for further investigation in this area. 

### Limitations of the Study

Establishing a causal relationship between an intervention and an outcome is a complex task [52]. A sufficient sample size is required to have enough statistical power to detect an effect, and small sample sizes make multivariate analyses impractical [53]. However, in the context of animal-assisted education (AAE) interventions, which are influenced by various factors, it is uncertain whether establishing a causal relationship is feasible. Using a control group is a methodologically justifiable option [52] to ensure that any changes in the outcome can be attributed to the intervention and did not merely occur naturally over time. Nonetheless, controlling and describing all the details was challenging due to the complexity of the population and interventions and the potential interplay between these factors. Additionally, according to Lutwack-Bloom, Wijewickrama, and Smith [54], the Hawthorne effect, where participants achieve better results due to the attention they receive or the novelty of the situation, should be taken into consideration. The outcomes of our research suggested that the positive effects of dogs on social and emotional outcomes may have been short-term, occurring during and shortly after the interaction. It is uncertain whether the results of our study had a lasting effect. Further research is needed. Regarding the observations, caution should be exercised when generalizing the outcomes to larger groups. We constructed an observation checklist and ensured inter-rater reliability. 

This study had several limitations. As an exploratory initial study, its scope was limited to assessing the positive impact of the DOG Project. The survey specifically focused on the influence of the 10 sessions of the DOG Project on the social and emotional outcomes of elementary school students aged 8 to 13 years. However, the study did not include a separate group to evaluate the program without a dog. The inclusion of such a group would have required additional efforts from the three participating dogs, potentially impacting their well-being and introducing a negative effect on the size of the experimental group. Moreover, the DOG Project relied on only one teacher/handler, and there were no other professionally trained individuals involved in the program. This limitation further affected the size of the experimental group.

This exploratory study had some acknowledged weaknesses, including the varying environmental conditions in schools, teacher age and experience, and diverse home situations. However, the study’s reliability was increased by the fact that the DOG Project was implemented by the same teacher/handler consistently. While a causal explanation between the intervention and social emotional development could not be provided, the intervention was promising. While both studies indicated a positive effect of the DOG Project, it is important to note that the sample sizes in these studies were small. Therefore, it is recommended to conduct follow-up studies with larger sample sizes. 

Additionally, it is advised to conduct further research on the motivational aspect of AAE. Such research could make use of the results of this exploratory study and the underlying indicators and involve a larger sample of students.

## 5. Conclusions

Hypothesis 1 stated that AAE positively impacts the social and emotional outcomes of elementary school students aged between 8 and 13 years, as reported by the students themselves after the intervention (self-confidence, work attitude, relationship with the teacher, and relationships with other students).

Based on the results, it appeared that the DOG Project had a positive impact on social-emotional development and social behavior in the short term, as reported by students. The post-test scores of students in the experimental group showed a significant increase in self-confidence and a more positive relationship with their peers, as reported by the students themselves. The self-confidence dimension indicated an improvement in their belief in their abilities to complete tasks, as well as a change in their self-image. The ‘relationship with other students’ dimension revealed an increase in liking for classmates, better communication with peers, and an enhanced ability to recognize and address undesirable behavior among peers. Accordingly, we could conclude that H1 was largely confirmed.

Hypothesis 2 stated that AAE positively impacts the social and emotional outcomes of elementary school students aged 8–13 years, as reported by their teachers after the intervention (working attitude, pleasant behavior, emotional stability, and social behavior).

Comparing the pre-test results to those of the post-test, the teachers rated students in the experimental group significantly higher on all four dimensions—attitude toward work, pleasant behavior, emotional stability, and social behavior. They perceived an improvement in the students’ attitude toward work and behavior toward peers and teachers, and they also believed that students became more considerate. The score also indicated that teachers had more confidence in the emotional stability of the students in the experimental group. The teachers observed that the social behavior of the students was more focused on participating in group activities and interacting with both the teacher and classmates. Thus, we could conclude that H2 was confirmed.

Hypothesis 3 stated that, underpinning H1 and H2, AAE positively impacts the level of intensity and energy reflected in the verbal and non-verbal communication of students during the intervention. 

Based on the analysis of the observed video material, it was evident that the AAE intervention had a positive influence on the intensity of both verbal and non-verbal communication. Several behavioral aspects showed significant improvements, with average scores increasing in different patterns between the free situation and tasks 1 and 2. The voice volume and articulation improved, making communication clearer and more understandable. The level of intensity and interest displayed by participants in the assigned tasks increased, with some even exhibiting hyperactivity. Additionally, physical contact with the dog showed a noticeable increase in conscious touch and more interaction with the dog. Table 3 shows that in some instances, the expected gradual increase in the intensity of behavior between the free situation, task 1, and task 2 was observed. In other cases, the pattern was more diffuse. Accordingly, we concluded that H3 was to some extent confirmed.

The conclusions emerging from this exploratory study seemed to support that AAE with dogs contributed to the positive development of students. This is consistent with other studies regarding social-emotional functioning [30,31]. The intervention seemed to produce a significant improvement in social-emotional aspects in the experimental group, whereas this was not the case in the control group. This improvement seemed to be related to an increase in the number of observable student behaviors that could be seen as drivers or precursors of social-emotional changes. Accordingly, the results of the questionnaire and observations complemented each other. Verheggen et al. [11] indicated mutual embodied attunement in behavior and emotion as a basis for theoretical frameworks such as attachment, social cognition, social support, and physiological effects [11]. Thus, we might conclude that this specific AAE intervention underpinned the idea of a positive impact between humans and dogs and the mechanism of enactive anthrozoology.

In conclusion, AAE is an expanding field of interventions in education. The current study indicates future directions for theoretical underpinnings, improved understanding, and the empirical measurement of underlying variables and mechanisms.

## Figures and Tables

**Figure 1 children-10-01316-f001:**
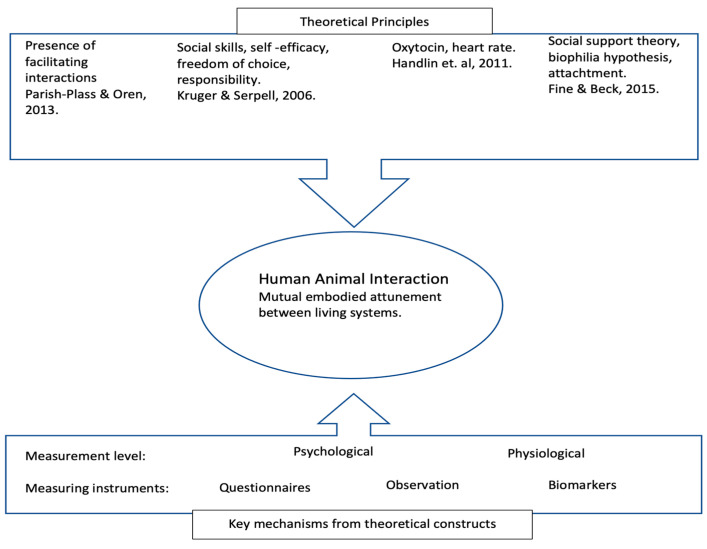
Based on Verheggen et al., 2017.

**Table 1 children-10-01316-t001:** Comparison of pre- and post-test means for experimental group (N = 22) as rated by the teachers and the students using VISEON.

	Pre-Test Mean	Post-Test Mean
** *Student self-assessment* **		
Self-confidence	2.57	2.76 *
Relationships with other students	3.02	3.20 *
Relationship with the teacher	3.53	3.55
** *Teacher assessment* **		
Attitude to work	2.81	3.07 **
Pleasant behavior	3.23	3.46 **
Emotional stability	2.31	2.83 **
Social behavior	2.25	2.74 **

*Note:* * *p* < 0.05, ** *p* < 0.01 based on Wilcoxon signed-rank test (one-sided).

**Table 2 children-10-01316-t002:** Comparison of pre- and post-test means for control group (N = 23) as rated by the teachers and the students using VISEON.

	Pre-Test Mean	Post-Test Mean
** *Student self-assessment* **		
Self-confidence	2.89	2.87
Relationships with other students	3.41	3.35
Relationship with the teacher	3.56	3.57
** *Teacher assessment* **		
Attitude to work	3.60	3.43
Pleasant behavior	3.62	3.43
Emotional stability	3.24	3.05
Social behavior	3.12	3.00

*Note:* based on Wilcoxon signed-rank test (one-sided).

**Table 3 children-10-01316-t003:** Mean differences in observation scales for communication during interactions between situations.

Communication Type		Mean Difference Free Situation—Task 1	Mean Difference Free Situation—Task 2	Mean Difference Task 1—Task 2
*Verbal*	Voice volume	1.33 **	1.83 **	0.51 **
	Articulation	1.30 *	1.78 **	0.48 **
*Non-Verbal*	Facial expression	1.56 **	1.28 *	−0.29
	Eye contact with trainer	0.21	−0.48	−0.71
	Eye contact with dog	0.18	0.12	−0.06
	Posture	1.69 **	1.65 **	−0.04
	Intensity	1.76 **	0.88 *	−0.88 **
	Contact with toy and dog	2.38 **	2.45 **	0.05
	Physical contact with dog	1.12 *	1.25 *	0.13

*Note:* * *p* < 0.05, ** *p* < 0.01 based on Wilcoxon signed-rank test (one-sided).

## Data Availability

The data supporting this study’s findings are available from the corresponding author upon reasonable request.

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
