# Peer review of "Animal-Assisted Education: Exploratory Research on the Positive Impact of Dogs on Behavioral and Emotional Outcomes of Elementary School Students"

_children, 2023, doi:10.3390/children10081316_

Round 1

Reviewer 1 Report

Dear Authors,

Thank you for your article. It investigated the effect of a canine assisted intervention programme in the school environment on the social and emotional development of children aged 8 -13 years old. Your introduction was interesting and well supported by literature. Some aspects of your research design and reporting were good, and avoided some of the typical problems encountered in projects of this type. For example, your hypotheses were explicitly stated in the introduction. In addition, the inclusion criteria, number, and allocation of participants were clearly described. The number of dogs involved was stated and their relationship to the handler/therapist was disclosed. Your ethical approval was provided at the end of the paper. The VISEON questionnaire was also well explained. It was novel and beneficial to use an outcomes measurement tool that employs self-assessment by the child, in addition to assessment by the teacher. However, I also have some serious concerns about the research design and reporting (specifically the control condition, animal welfare, poor reporting of the intervention, and your statistical analyses).

One concern is that although you used a control group in the study, the children in the group remained in their classroom and received no intervention. Therefore, it is impossible to tell whether the measured improvements were caused by the dog or could be attributable to other aspects such as attention from the therapist, or the activities and conversation involved. Ideally, the children in the control group should have some time or activities with the therapist but without the dog.

I am also concerned about the welfare of the dogs involved. If each of the 28 children received 10 sessions, then each dog would have been involved in 93 sessions. This seems excessive. How long was each session and how were the visits spaced out for each dog? You stated that the handler (owner) was trained and monitored the welfare of the dogs. This would be difficult for them to do while also delivering other aspects of the intervention. What exactly was monitored-behaviour? If so, how and what indicators were used? What would prompt withdrawal of the dog and did this happen during any of the sessions? You provided the number and breed of the dogs and stated that they were heathy and vaccinated. Details such as their sex and neuter status, age, and husbandry (including diet and anthelmintic treatments) were omitted. Were students educated on safe child-dog interactions prior to starting the sessions? The IAHAIO provides guidelines for safeguarding human and animal welfare during animal assisted interventions. You have actually cited and referenced this document in your paper. They state that it is important to avoid certain interactions during sessions (jumping, bending over, dressing up, teasing, pulling on body etc.). Was this adhered to?

In addition to this, your reporting and statistical analyses need improvement. What actually happened during each of the 10 sessions? Only a basic overview was given. The protocol should be moved into the methodology rather than being presented as Appendix B. It also needs to be expanded on. It may also be helpful to include a table summarising the specific activities carried out during each session 1-10. How was the coding done? This was not described. What software and version number did you use for your statistical analysis? This needs to be reported. You need to statistically compare the results from your experimental and control groups.

Be cautious about your interpretation of the videos in the observational study. A loud voice may indicate anger or distress, not improved social interactions or emotional regulation. Eye contact with a dog may not be appropriate. Many dogs are uncomfortable with eye contact and some children may understand this. High intensity may indicate anxiety/fear or hyperactivity which would not be an improvement in the child.

Research in human-animal interactions is notoriously challenging to design, due in part to the large number of confounding factors involved. I would like to draw your attention to an excellent paper that highlights this and may be helpful to you: Stern, C. and Chur-Hansen, A. (2013) ‘Methodological Considerations in Designing and Evaluating Animal-Assisted Interventions.’ Animals, 3, pp.127-14. [Online] Available at: https://doi.org/10.3390/ani3010127

Please also see some specific line-by-line comments below:

Specific comments:

1.    Line 15 (Abstract):

Remove your reference to ‘academic outcomes’, as academic outcomes were not evaluated.

2.    Lines 16-20 (Abstract):

Clarify that the significant differences were not from group wise comparison.

3.    Lines 30-31 (Introduction):

Requires citation to support your point.

4.    Lines 33-35 (Introduction):

Requires citation to support your point.

5.    Lines 142-150 (Introduction):

Remove references to ‘academic outcomes’, as academic outcomes were not evaluated.

6.    Introduction (General):

This will be more relevant if you fully describe the activities involved in each visit/intervention. Did they involve structured play? Guided physical contact/petting? Observation and talking about the emotions of the dog?

7.    Line 160 (Procedure):

Academic functioning was not assessed.

8.    Lines 166-169 (Procedure):

Did the children also communicate their assent for participation?

9.    Lines 172-173 (Study Design):

Without a control treatment or exposure to the handler/therapist alone this is not the case.

10. Lines 176-177 & 179-182 (Study Design):

What sort of help needed to be requested by the school and how did they request such help?

11. Lines 194-195 (Study Design):

Do you mean that they filled out questionnaires both before and after each session or at the start and the end of the intervention programme?

12. Lines 242-243 (Study Design):

Requires citation to support your point.

13. Line 258 onwards:

Add sections describing the characteristics of the dogs, the interventions (1-10), and the monitoring of animal welfare. Add a table summarising the interventions.

14. Lines 260-261 (Analysis):

How was the coding performed? What software (and version) was used for the statistical analyses?

15. Lines 273-274 (Results questionnaires):

The separate pre-post analysis of the experimental and control groups was not appropriate.

16. Lines 301-303 (Study Design):

Please see my previous comments on the video analysis.

17. Lines 370-372 (Discussion):

You cannot draw this conclusion as a control treatment was not used and your statistical analysis was inappropriate.

18. Line 381 (Discussion):

Academic outcomes were not assessed.

19. Line 392 (Discussion):

You cannot draw this conclusion as a control treatment was not used and your statistical analysis was inappropriate.

20. Lines 469-471 (Discussion):

Discuss the limitations more thoroughly.

The English language used was very good.

Reviewer 2 Report

Dear Authors, I was pleased to review your manuscript which deals with such a delicate topic in the field of Animal Assisted Interventions.

Here are my comments and suggestions:

The Introduction section is too long, it would be advisable to try to summarize it.

The experimental design has been designed in an articulated and detailed manner but, in this regard, there are some points that I would prefer to be clarified: - perhaps it will have escaped me but I have not found a clear reference to the period in which the meetings with the children took place ;

- the method of recruiting children from both the experimental and control groups should be better explained;

- the age range of children aged 8-13 does not include their placement in elementary schools worldwide. Please specify that this is the case in the Netherlands;

- it is reported that 3 dogs were involved, but of these, we have no generic characteristics or other assessments of temperament or behavior. It would be appropriate to integrate.

- was there a dog handler or were the activities carried out by the teacher? It is not clear in the text;

- please give more details of the video rating, as the cited reference is very broad and subjective;

The Conclusions are not mandatory but they certainly shouldn't be merged with the Discussion into a single section. Please split.

Bibliographic references in the text have not been reported as indicated in the template. Please modify.

Author Response

Dear esteemed reviewer,

We would like to express our gratitude for taking the time to review our article. Your expertise, insights, and constructive feedback improved the quality of our article. Your careful evaluation and thoughtful comments have undoubtedly enriched our article and helped us refine our ideas. We recognize the effort and expertise required to provide a thorough review, and we are grateful for your dedication to maintaining the highest academic standards.

Please be assured that we have carefully considered each of your comments and suggestions, and we have made every effort to address them appropriately in our revised article.

With profound gratitude,

Riki Verhoeven

University of Applied Sciences, Utrecht; Open University, Heerlen.

Reviewer 3 Report

A brief summary

The study idea is good, but the manuscript needs arrangement for the headings with gathering all the methods or tools used under the same heading and the same for results. The objective cleared from the study is proof of the positive effect of animal assisted education mentioned in the hypothesis not to determine the impact of animal assisted education as mentioned which need to be revised together with title of the manuscript.

Specific comments

Title:

-         Add (positive) before (impact), add ( of school students) after (outcomes)

Abstract

-         line 13: no reference in the abstract so remove

-         line 14-15: add (in children) after (development), remove (reports…………..to), add (through mixed method) after (years)

-         N.B. add the methods used in brief before results here in the abstract

-         Line 23-24: not clear meaning, where is the conclusion from the study

Introduction

-         Line 55: add year to the reference

-         Line 83-84: remove (the), remove “ “ of playful

-         Line 85-86: remove (such………..features), remove (thus)

-         Line 88: write (were) not (we), add the number of studies reviewed.

-         Line 93: write (emotional) not (emotion)

-         Line 95-96: write (features) not (factors)

-         Line 110: rewrite to: According to Correale et al. (2017), elementary school teachers perception is that…………….

-         Line 114: not clear its meaning, clarify

-         Line 125-134: not flow with lines 121-124

-         Line 128-130: not clear, clarify

-         Line 143-153: all hypothesis mentioned are for the positive impact of AAE while the main objective or question of gap of knowledge mentioned is Does AAE impact the social, emotional 141 and academic outcomes of elementary school students aged between 8–13 years?

Survey

-         Line 156: Driegang??? Add more information about it which is present in appendix b which can be moved here for more clarification

-         Line 157: is the teacher or handler in the intervention are the teacher of the classroom??

-         Line 183: the 37 students is for both groups or the same number in each group ?

-         Line 186-188: how remain the student remain anonymous to teacher and help him/her in the questionnaire and from its classroom, is the number of teachers is the same as students??

-         Line 188-190: teachers not respond to what ?? clarify , 28 from each group not from both groups?? Clarify

-         Line 188-193: the change in the number of students is before or after intervention? Clarify

-         Line 195: mention the meaning of VISEON

-         Line 197-199: not understandable , clarify

-         N.B. move the study design part before the procedure part in both studies, add the ethical approval

-         Line 213: 44 pairs means 88 questions?? Is it a scale questions? Clarify

-         Line 252-254: move before line 234, mention the scale used for both students and teachers questionnaire

-         Line 262: what you mean by semi continuous variable

-         Line 273-274: unclear explanation for separate analysis for both groups and the result for the control group is logic and not need a pre and post test comparison as they have already no intervention exposure. The comparison between groups is important in the post test condition to clarify the difference between groups which is similar to what you have done for the pre post test of experimental group., so why use control group?? Also it is not mentioned in the methods that the questionnaire is pre and post intervention.

Observational

-         Line 296: mention the length of video analyzed

-         Line 298: mention the number of schools

-         Line 308-310: clarify with mention of video length used, mention the score used for measures coded form the videos

-         Line 312-319: is not measures?? edit heading

-         Line 321-332: rewrite its heading to video analysis not analysis

-         Line 334-339: move to video analysis which is the previous heading instead of analysis

-         Line 340-347: give it a heading of statistical analysis

-         Line 342: mean difference for what? Clarify

-         Line 350-362: give it a heading of results, also you don`t mention the sex used in the methods

Discussion

-         Line 368-369: remove

-         Line 457: mention studies

English is good except some minor editing. 

Author Response

Dear esteemed reviewer,

We would like to express our gratitude for taking the time to review our article. Your expertise, insights, and constructive feedback improved the quality of our article. Your careful evaluation and thoughtful comments have undoubtedly enriched our article and helped us refine our ideas. We recognize the effort and expertise required to provide a thorough review, and we are grateful for your dedication to maintaining the highest academic standards.

Please be assured that we have carefully considered each of your comments and suggestions, and we have made every effort to address them appropriately in our revised article. Your feedback has strengthened our research.

With profound gratitude,

Riki Verhoeven

University of Applied Sciences, Utrecht; Open University, Heerlen.

Round 2

Reviewer 1 Report

Dear Authors,

Thank you for giving my comments due consideration and making relevant adjustments to your manuscript.

I am glad that you have now included a table detailing the activities involved in each session. This will enable the reader to better understand what was done and how the activities may have affected the dogs. I agree that trying to determine the most beneficial aspects of the sessions would be prohibitively complex. I actually think that it would be interesting to read a separate paper on how you designed the programme and sessions, as this is rarely reported in the literature and would be helpful to others. The sessions appear to be designed to improve children's focus, confidence, and autonomy. In addition to teaching them how their behaviour/body language and voice may affect others, to understand the perspectives of others, and how to be a good leader. Is this the case?

In addition, thank you for adding some more details about the safeguarding of canine welfare. However, I still feel that more are needed. It would be an error to assume that interactions with positive intentions, such as petting, play, and positive training, are always enjoyed by an animal. Petting in an inappropriate way or for prolonged periods may be stressful. Frustration, over arousal, fatigue, or anxiety may even occur during play or positive training. From your comments, I presume that the handler was monitoring canine stress using behavioural parameters, rather than using physiological stress indicators or a combination of both? It would be good practice to detail the specific behaviours that were monitored for and that would have caused the handler to withdraw the dog and end the session.

I still fundamentally disagree with your control condition and your statistical analysis. You note that special attention alone may be beneficial and you did not want to compare the involvement of a dog to a different intervention. However, you are reporting on the efficacy of a canine assisted intervention, so it is essential to show that the dog was indeed necessary. This is also important from an ethical perspective and the '3Rs' of animal use- reduction, refinement, replacement. Why involve a dog rather than a person or another activity? To avoid the equivalent of a 'placebo effect', group wise statistical comparisons are needed, even with small sample sizes and lack of randomisation.

Very good.

Reviewer 3 Report

A brief summary

The manuscript is so much improved and easily readable except some few comments and arrangements below.

Specific comments

Abstract

-         Line 15: write (positively) before (impacts)

-         Line 25-26: write conclusion for the whole work before these lines, also this lines doesn`t make sense

Introduction

-         Line 74: write (of children’s) before (in education), add (as) before (it)

-         Line 76: write (was) before (conducted)

-         Line 86: write (feature) not (factor)

-         Line 108: write (although of this) before (there)

-         Line 109: write (example) not (option)

-         Line 122: write (positively) before (impact)

Survey

-         Line 159: clarify, (there had to be a request for help)?

-         Line 174: write the number of ethical approval

-         Line 223-225: repeatable, remove

-         Line 245-247: move before line 238-244

-         Line 263-268: move to procedure section before line 201

-         Line 269-287: move to procedure section before line 208

-         Line 273: clarify

-         Line 276: write (the) not (her)

-         Line 278: is the sessions occur in daily basis, clarify

-         Line 263-287: move to procedure section or put under separate heading before outcome measures

-         Line 303-306: move to study design section

-         Line 308: remove (small sample size) as it not make sense as not prevent comparison

-         Line 319-322: write that it is a comparison with the pretest

-         Line 321-322: it was nonsignificant ? revise

-         N.B. you can still compare the two groups from both tables, mention.

Observational

-         N.B. are these students the same as in the survey study or different ? clarify, are also those in the survey from different schools? Clarify

-         Line 332: mention the total length of session

-         Line 336-349: move to video analysis section

-         Line 352: there is some difference in the procedure. Clarify

-         Line 354-361: move to study design section

-         Line 384-395: put under separate heading of statistical analysis

Conclusion

-         Line 413: students only not teachers reports

-         N.B. move this part after discussion section

Discussion

-         Line 462: remove (in conclusion)

-         Line 478: remove (that there is)

-         Line 513 to end: put under separate heading of limitation of the study

-         Line 539-546: remove, irrelevant

-         Line 550-552: move to conclusion end and abstract

Minor English editing required

Round 3

Reviewer 1 Report

I still do not agree with the experimental design and would like more reporting on the process of monitoring animal welfare. However, I am satisfied with the modifications made to the manuscript.

Very good. There are a couple of 'typos' present.